

# Storage media and RNA extraction approaches substantially influence the recovery and integrity of livestock fecal microbial RNA

Raju Koorakula[1,2], Mahdi Ghanbari[3], Matteo Schiavinato[4], Gertrude Wegl[3], Juliane C. Dohm[4] and Konrad J. Domig[1]

[1] University of Natural Resources and Life Sciences, Vienna, Department of Food Science and Technology, Institute of Food Science, Vienna, Austria
[2] Competence Centre for Feed and Food Quality, Safety and Innovation (FFoQSI), Tulln an der Donau, Lower Austria, Austria
[3] DSM - BIOMIN Research Center, Tulln, Austria
[4] University of Natural Resources and Life Sciences, Vienna, Department of Biotechnology, Institute of Computational Biology, Vienna, Austria

Corresponding author
Mahdi Ghanbari,
mahdi.ghanbari@dsm.com

## ABSTRACT

**Background:** There is growing interest in understanding gut microbiome dynamics, to increase the sustainability of livestock production systems and to better understand the dynamics that regulate antibiotic resistance genes (*i.e.*, the resistome). High-throughput sequencing of RNA transcripts (RNA-seq) from microbial communities (metatranscriptome) allows an unprecedented opportunity to analyze the functional and taxonomical dynamics of the expressed microbiome and emerges as a highly informative approach. However, the isolation and preservation of high-quality RNA from livestock fecal samples remains highly challenging. This study aimed to determine the impact of the various sample storage and RNA extraction strategies on the recovery and integrity of microbial RNA extracted from selected livestock (chicken and pig) fecal samples.

**Methods:** Fecal samples from pigs and chicken were collected from conventional slaughterhouses. Two different storage buffers were used at two different storage temperatures. The extraction of total RNA was done using four different commercially available kits and RNA integrity/quality and concentration were measured using a Bioanalyzer 2100 system with RNA 6000 Nano kit (Agilent, Santa Clara, CA, USA). In addition, RT-qPCR was used to assess bacterial RNA quality and the level of host RNA contamination.

**Results:** The quantity and quality of RNA differed by sample type (*i.e.*, either pig or chicken) and most significantly by the extraction kit, with differences in the extraction method resulting in the least variability in pig feces samples and the most variability in chicken feces. Considering a tradeoff between the RNA yield and the RNA integrity and at the same time minimizing the amount of host RNA in the sample, a combination of storing the fecal samples in RNALater at either 4 °C (for 24 h) or −80 °C (up to 2 weeks) with extraction with PM kit (RNEasy Power Microbiome Kit) had the best performance for both chicken and pig samples.

**Conclusion:** Our findings provided a further emphasis on using a consistent methodology for sample storage, duration as well as a compatible RNA extraction

approach. This is crucial as the impact of these technical steps can be potentially large compared with the real biological variability to be explained in microbiome and resistome studies.

# INTRODUCTION

Gut microbial communities play important multifactorial roles in animal physiology. They are important in controlling pathogen colonization and in immune system development. They also help in digestion by metabolizing the compounds in diet, which could not be broken down through enzyme production by the animal host, and in the production of vitamins, such as vitamins B12, B5, and K (*Ikeda-Ohtsubo et al., 2018*).

Characterizing the livestock gut microbiota in terms of taxonomy and phylogeny has been carried out in a large number of studies by sequencing of the 16S ribosomal RNA subunit gene, of which either the full length or its hypervariable regions are targeted for sequencing and used for taxonomic classification (*Deusch et al., 2015*; *Ikeda-Ohtsubo et al., 2018*). Shotgun metagenomic sequencing adds a more detailed insight to the taxonomical characterization of a sample, by providing information about functional and genetic microbiome variability (*Wooley, Godzik & Friedberg, 2010*). Yet, it does not distinguish whether this information comes from cells that are viable or not or whether the predicted genes are actually expressed (*Knight et al., 2018*). In fact, microbial encoded genes are not necessarily matching their transcription (*Gallardo-Becerra et al., 2020*). A study conducted in the human gut microbiome by *Franzosa et al. (2014)* observed that up to 41% of microbial transcripts have different relative abundances when compared to their genome content. To overcome that limitation, high-throughput sequencing of RNA transcripts (RNA-seq) from microbial communities (metatranscriptome) allows an unprecedented opportunity to analyze the functional and taxonomical dynamics of the expressed microbiome (*Bikel et al., 2015*). Moreover, metatranscriptomics (MTX) is used as an effective method to assess which antibiotic resistance genes (ARGs) are actively transcribed by the microbiome (*Marcelino et al., 2019*). When appropriately applied in combination with metagenomics, it clearly unfolds which of the genes that were annotated in the metagenomic analysis are transcribed and to what extent, giving further insight into the physiological functions from a potential repertoire of bacteria that are actually active in a given context (*Bashiardes, Zilberman-Schapira & Elinav, 2016*; *Bikel et al., 2015*; *Franzosa et al., 2014*; *Knight et al., 2018*).

While there are benefits of using MTX to assess the gut microbiome, experimental bias can be introduced during critical experimental steps (*Bashiardes, Zilberman-Schapira & Elinav, 2016*; *Knight et al., 2018*). These key steps include: selection of sample storage media, temperature and length of storage, the RNA extraction method, *etc.* (*Bashiardes, Zilberman-Schapira & Elinav, 2016*; *Franzosa et al., 2014*; *Knight et al., 2018*). Information
about the effect of the experimental bias on the livestock gut microbiome is essential for large-scale, time-series and field microbiome analysis projects. Maintaining sample integrity constitutes a special concern in these situations, as it is not logistically feasible for researchers to collect and process samples on the same day and in many cases freezing at −80 °C (considered as the gold standard for microbiome material) is not possible (*Peimbert & Alcaraz, 2016*; *Song et al., 2016*). Instead, samples are usually taken at various time points and are collected and stored for future analysis. Next to the limitation of obtaining high quality and sufficient quantity of bacterial RNA, coextraction of host RNA is also an important consideration (*Bashiardes, Zilberman-Schapira & Elinav, 2016*; *Bikel et al., 2015*). Thus, a good method for sample storage, including both storage media and length of time of storage, and a subsequent compatible RNA extraction protocol is essential for downstream next generation sequencing to accurately recover the gut microbiome.

Our primary goal in this study was to determine the impact of the sample storage and RNA extraction strategies on the recovery of high-quality microbial RNA from selected livestock (chicken and pig) fecal samples. These sample matrices were chosen because the vast majority of livestock microbiome efforts are performed on fecal samples. We selected two currently used storage media RNALater stabilization reagent (referred to as RL) and DNA/RNA Shield™ (referred to as ZM) used in two storage conditions (overnight at 4 °C or 2 weeks at −80 °C), and in combination with four commercially available microbial RNA extraction kits, the QIAGEN RNeasy Power Microbiome total RNA Kit (referred to as PM) (QIAGEN, Hilden, Germany), the Norgen Biotek stool RNA Kit (referred to as NO) (Norgen Biotek Corp, Thorold, ON, Canada), the ZymoBIOMICS RNA miniprep Kit (referred to as ZY) (Thermo Fisher Scientific, Waltham, MA), and the Macherey-Nagel NucleoSpin RNA stool Kit (referred to as MN) (Machery-Nagel Fisher Scientific, Düren, Germany). We evaluated the performance of these approaches using metrics that have been previously shown to be affected by the storage media and extraction method including RNA yield, purity and more importantly RNA integrity (*Reck et al., 2015*).

## MATERIALS AND METHODS

### Sample collection and stabilization

Fecal samples from a 180 days old pig ($n = 1$) and 55 days old chicken ($n = 8$) were obtained from conventional slaughterhouses. Subsequent to the euthanasia of the animals, the gastrointestinal tract was removed, stored on normal ice (4 °C) and immediately transported to the laboratory (~15 min transportation time). Samples from the digesta in the distal part of the large intestine, herein referred to as feces, were gathered. The pig fecal sample, as well as the pooled chicken fecal samples, were homogenized and then transferred into the tubes containing the defined storage media including RNALater (QIAGEN GmbH, Hilden, Germany) stabilization reagent (RL) and DNA/RNA Shield™ (Zymo Research, Irvine, CA, USA) reagent (ZM). After an overnight storage at 4 °C, to mimic the real field and transportation condition, half of the tubes in each storage media were immediately subjected to the RNA extraction procedure, while the other half was stored at −80 °C for 2 weeks until extraction. The duration of the freezing period was

considered of minor importance in this study and was chosen to reflect realistic conditions based on in-house work routine (*Seelenfreund et al., 2014*).

## RNA extraction

Extraction of total RNA was done using four different commercially available kits, namely the RNeasy PowerMicrobiome kit, (QIAGEN GmbH, Hilden, Germany), the Stool Total RNA purification kit (Norgen Biotek crop. Thorold, Canada), ZymoBIOMICS™ RNA Miniprep kit (Zymo Research, Irvine, CA, USA), and the NucleoSpin RNA stool kit (Macherey-Nagel GmbH, Düren, Germany). The RNA extraction was conducted in triplicate, yielding twelve samples (*i.e.*, three per combination) per each animal type. The combinations were the following: RL stabilization + short storage (4 °C, 24 h); RL stabilization + long storage (4 °C, 24 h followed by −80 °C, 2 weeks); ZM stabilization + short storage; ZM stabilization + long storage. The tubes containing samples were thawed on ice and then vortexed for homogenization. The content of each tube was transferred into individual 2 ml Eppendorf tubes and centrifuged with 15,000 × g (Eppendorf Centrifuge 5424 R) for 5 min at room temperature (25 °C) to pellet out the feces and to remove the supernatant, which could contain traces of stabilizers. The RNA extraction from all samples was carried out following kit protocols with minor adjustments: Firstly, 150 mg of sample were used as initial starting material, and secondly, 80 μl of RNAse-free water was used instead of 100 μl for the elution of all the extracted RNA.

## RNA extraction using the QIAGEN RNeasy Power Microbiome total RNA kit (PM)

Total RNA isolation was performed according to the manufacturer's instructions.

## RNA extraction using the Norgen Biotek stool RNA kit (NO)

Total RNA extraction was carried out as per the manufacturer's protocols.

## RNA extraction using the ZymoBIOMICS RNA miniprep kit (ZY)

Total RNA purification was performed according to the manufacturer's instructions.

## RNA extraction using the Macherey-Nagel NucleoSpin RNA stool kit (MN)

Total RNA isolation was performed according to the manufacturer's instructions.

## Determination of RNA concentration and integrity

Three methods were used to check the RNA quantity (concentration) and quality (purity and integrity) in each of the 96 samples. Firstly, the Nano-Drop spectrophotometer (Thermo Scientific, Wilmington, DE, USA) was used to check the absorbance at different wavelengths. The 260/280 ratio was used to estimate the purity of RNA regarding compounds absorbing UV light (*e.g.*, proteins). The 260/230 ratio was used to estimate the presence of contaminants (*e.g.*, salts). Samples with a 260/230 ratio in the range of 2.0–2.2 were considered to contain a sufficiently pure RNA. Secondly, the concentration and the quality of the total RNA was determined with the Qubit 4 fluorometer

(Thermo Scientific, Wilmington, DE, USA). The total RNA was quantified according to the Qubit RNA XR Assay Kit protocol. The RNA integrity and quality were quantified according to the Qubit RNA IQ Assay Kit protocol. Thirdly, the RNA integrity/quality and concentration were measured using a Bioanalyzer 2100 system (Agilent, Santa Clara, CA, USA) with capillary electrophoresis, using the RNA 6000 Nano kit (Agilent, Santa Clara, CA, USA). Bioanalyzer RNA 6000 Nano assay provides a measurement of RNA integrity (quality) quantified by the RNA Integrity Number (RIN). The RIN and concentration were checked following the manufacturer's protocol.

## Transcriptional analysis by RT-qPCR

To determine if the extracted RNA was functional, Reverse Transcription (RT-qPCR) was applied, using a previously established assay targeting the bacterial 16 S rRNA gene (*Liu et al., 2012*). To further evaluate the contamination of RNA extracts with host RNA, RT-qPCR targeting the chicken (*gallus gallus*, NM_001252255), or pig (*sus scrofa*, NM_001001636) ribosomal protein L32 (*RPL32*) gene was performed. Details on the qPCR assays (primer/probe sets, target genes, reagents and cycling conditions) are summarized in Table S2.

After assay optimization and evaluation in the corresponding matrices, an equivalent volume from all samples, three biological replicates per extraction method and host organism, were then assayed by qPCR in technical triplicates both with the BactQuant assay and the assays against the chicken and pig *RPL32* genes in two duplex reactions (The BactQuant assay combined with either the gallus RPL32 or the sus RPL32 assay; Table S2). RT-qPCR was conducted in a one-step reaction using a the SOLIScript 1-step Multiplex Probe Kit (Solis Biodyne, Tartu, Estonia). Each qPCR reaction composition consists of 10 µl reaction mixture and 2.5 µl of the isolate was used as template in a 12.5 µl one-step qPCR reaction. The resulting Ct values were converted into absolute values using the $2^{-\Delta Ct}$ formula. For each assay, a corresponding set of qPCR reactions lacking the reverse transcription step was performed as control for signal derived from DNA template. The results indicated that the DNA quantities present in the samples are less than 25% in all cases (delta Ct >2).

## Statistical analysis

For analysis of RNA yield and RNA integrity number (RIN), a linear regression was performed where the predictors were sample type, extraction method, storage media, storage temperature and an interaction term between the sample type, extraction method, storage media, storage temperature to determine if the final yield or quality of the extracted RNA was impacted by interaction of the variables. In case the interaction term had a *P*-value of <0.05, we performed subsequent linear regression stratified by sample type, followed by a pairwise analysis. The normality of the data was checked with Shapiro-Wilk-Test (*Shapiro & Wilk, 1965*) and where necessary the data was normalized based on the output from the best Normalize R package (*Peterson & Cavanaugh, 2020*).

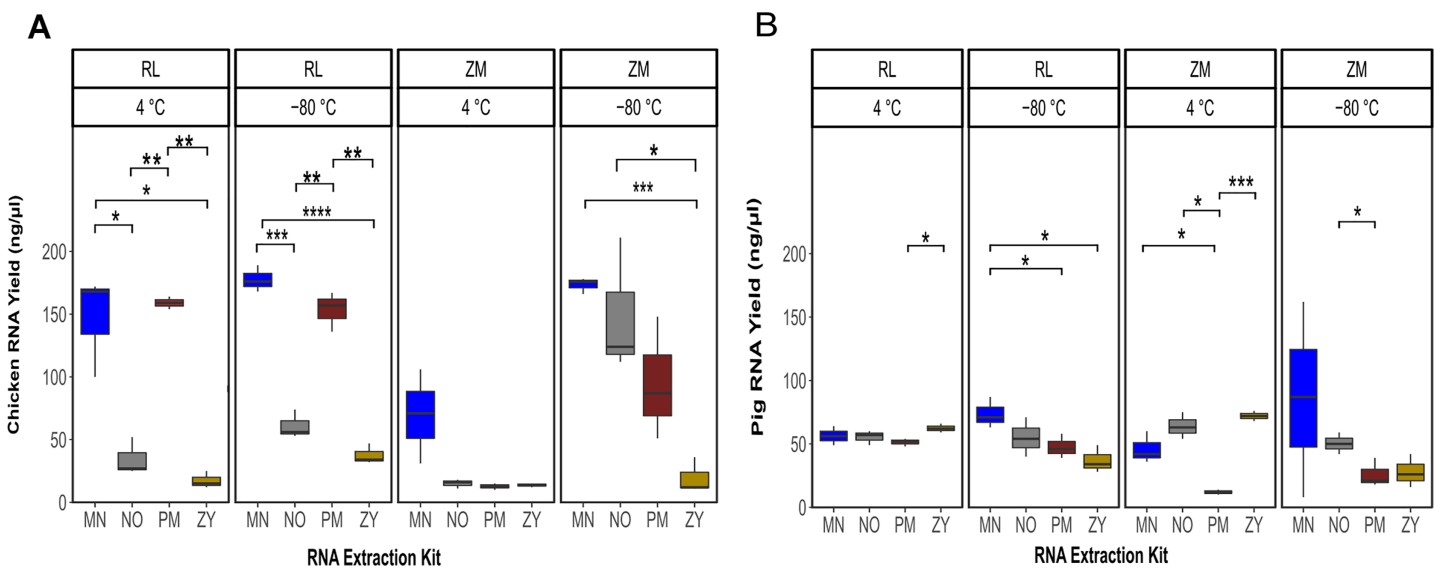

**Figure 1 Total RNA yield.** Total RNA yield obtained from chicken feces (A) and pig feces (B) using four different RNA extraction kits (MN, NO, PM, ZY), two different stabilization kits (RL and ZM), and two different storage temperatures (4 °C and −80 °C). Boxplots represent triplicate values, the median is indicated by a horizontal line within the box. Significant differences were tested with the Kruskal–Wallis paired t-test, *$P < 0.05$, **$P < 0.01$, ***$P < 0.001$, and ****$P < 0.0001$.

## RESULTS

### RNA extraction, storage and their interaction affect the recovery of RNA

Linear regression analysis showed a significant effect of the sample type (pig or species feces; $P = 0.02$), storage media ($P = 1.6e{-}07$), storage temperature ($P = 1.1e{-}05$) and the extraction kits ($P = 1.4e{-}11$) on the final RNA yield. The sum of the square value indicated that among the different experimental variable, the extraction kit accounted for the majority of the variability in the RNA yield (SSR = 12.50), followed by storage media (SSR = 5.22), storage temperature (SSR = 3.41) and the sample type (SSR = 0.78). In general, a significantly higher amount of RNA was recovered from the chicken feces samples compared to the pig fecal material (Fig. 1; Table S1). Of note, the interaction term between the sample type and other variables was not statistically significant ($P = 0.19$), suggesting that one specific combination of the storage media and storage temperature followed by a specific RNA extraction methodology resulted in a higher yield of RNA, regardless of the sample type. In fact, the highest yield of the RNA was obtained from the samples stored in RL for 2 weeks at −80 °C and extracted with the MN kit (Fig. 1). For the chicken feces samples, PM extraction kit showed the second-best performance in terms of DNA yield, achieving (significantly) higher RNA recovery from the fecal samples ($P_{PM\ vs\ MN} = 0.02$, $P_{PM\ vs\ NO} = 0.06$, $P_{PM\ vs\ ZY} = 3.17e{-}6$) after adjusting for the storage media and the storage time (Fig. 1A). Linear mixed effect model analysis with the RNA extraction methods as covariate revealed that sample storage duration was a predictor of RNA yield, with the samples stored at −80 °C associated with a significantly higher RNA concentration ($P = 1.68e{-}06$; Fig. 1). Of the investigated storage media,

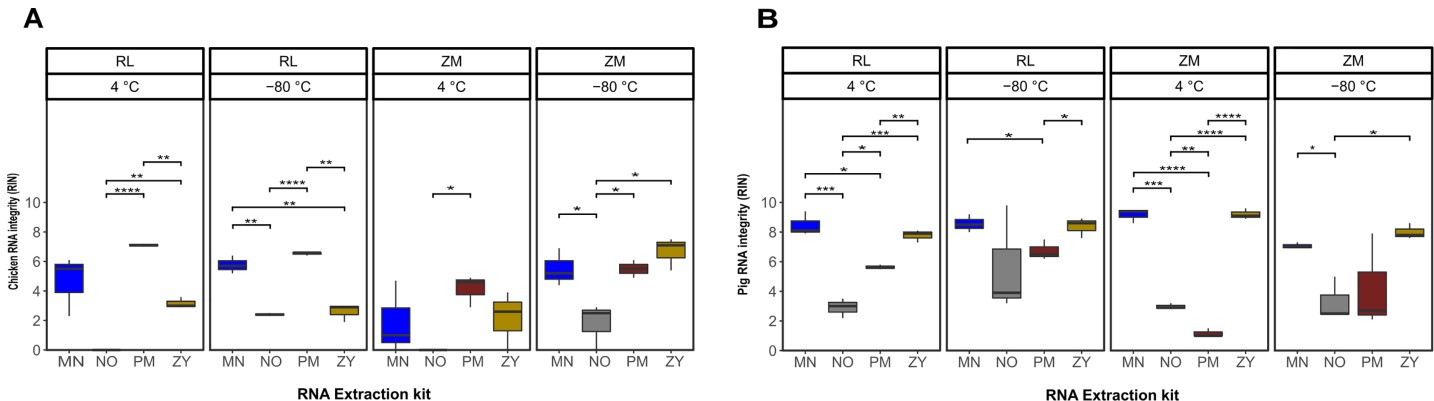

**Figure 2 Total RNA integrity.** Total RNA integrity in samples obtained from chicken feces (A) and pig feces (B) using four different RNA extraction kits (MN, NO, PM, ZY), two different stabilization kits (RL and ZM), and two different storage temperatures (4 °C and −80 °C). Boxplots represent triplicate values, the median is indicated by a horizontal line within the box. Significant differences were tested with the Kruskal–Wallis paired t-test, *$P < 0.05$, **$P < 0.01$, ***$P < 0.001$, and ****$P < 0.0001$.

storing the chicken feces in RNALater resulted in a higher recovery of RNA, regardless of the storage duration ($P = 0.0004$; Fig. 1A). Linear regression analysis of the RNA yield from the pig feces revealed that the type of the storage and the storage duration are not the predictor of the RNA yield but the RNA extraction methodology is ($P = 0.009$; Fig. 1B). Specifically, the interaction term between the storage time and the extraction kit was significant, indicating that there was no single extraction method that had the highest RNA yield for all storage media and storage temperature. For the pig feces samples stored at 4 °C, ZY, NO and MN comparatively resulted in higher yield of RNA, while PM extraction kit output was the lowest (Fig. 1A). For the pig feces samples stored at −80 °C for 2 weeks, generally the yield was higher compared to the one stored at 4 °C, though not significant. The MN extraction kit significantly performed better than the other three extraction methodologies, achieving higher RNA recovery from the pig feces samples ($P \leq 0.03$).

## RNA extraction, storage and their interaction affect the integrity of RNA

The statistical analysis of the RNA integrity data based on the RIN as response variable demonstrated that on average the RNA extracted from the pig feces had significantly higher integrity compared to the ones extracted from chicken samples ($P = 0.0002$). Regardless of the sample type, the combination of the storage in ZM buffer for 2 weeks at −80 °C and extraction using the ZY kit resulted in a higher integrity of the extracted RNA (Fig. 2; Table S1). Linear regression analysis showed that for both sample types, the extraction kit was the main predictor of the RNA integrity (sum of squares regression $SSR_{chicken} = 144.72$, $SSR_{pig} = 220.47$, $P < 0.001$). The second main predictor for the chicken fecal samples was the storage temperature ($SSR = 35.02$, $P = 0.0004$), while the second main driver of the RIN ($SSR = 15.07$, $P = 0.02$) for the pig feces samples was the storage media. The chicken samples stored at 4 °C, extracted by the PM kit showed significantly ($P \leq 0.01$) higher integrity of the RNA, regardless of the storage media, although sample storage in RL resulted in a higher RIN number. However, for the samples

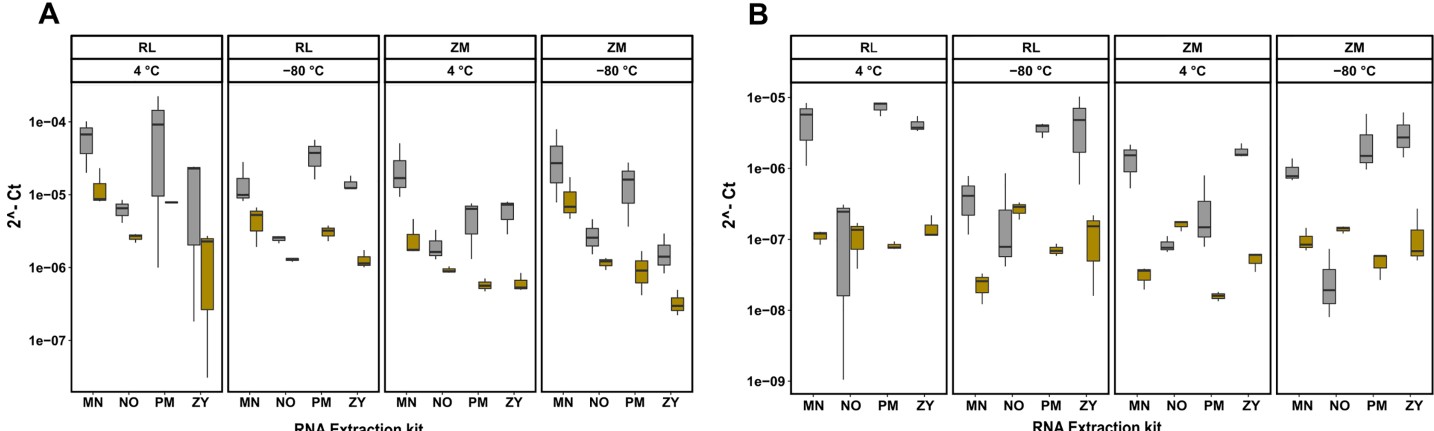

**Figure 3** **RT-qPCR data of 16S rRNA and RPL32 gene transcript RNA from all the extracted RNA samples.** Bacterial 16S and RPL32 gene transcripts from chicken and pig were measured by RT-qPCR in chicken feces (A) and pig feces (B) using four different RNA extraction kits (MN, NO, PM, ZY), two different stabilization kits (RL and ZM), and two different storage temperatures (4 °C and −80 °C); In box plots, gray and yellow colored bars represent 16s rRNA and RPL32 gene transcripts, respectively.

stored at −80 °C, a storage media dependent effect was observed, mainly for the ZY kit (Fig. 2A). For example, the RNA extracted by the ZY extraction kit from the sample stored in ZM buffer, showed the highest RNA integrity based on RIN number. However, the same kit resulted in a very poor integrity of RNA extracted from the chicken samples stored in RL buffer (Fig. 2A). On average, the integrity of the RNA extracted from the pig feces samples stored in RL buffer was significantly higher than the ones stored at ZM buffer ($P = 0.014$). Regardless of the storage media the MN and ZY extraction kits demonstrated a good performance in terms of the integrity of extracted RNA (Fig. 2B).

## RT-qPCR analysis confirm coextraction of host RNA

Next to quantity, purity and integrity, RNA extracts were further evaluated for functionality by detecting the bacterial 16S rRNA gene. To further quantify the extent of coextraction of eukaryotic RNA, thereby adding an additional quality metric allowing to select conditions with high amounts of bacterial RNA, the chicken, respectively swine *RPL32* gene transcript was assessed in downstream RT-qPCR analysis. Amplification was possible for all samples, revealing absence of potential PCR inhibitors in the extracts. The RT-qPCR results presented in Fig. 3 show differences between extraction kits and storage conditions for both targets and reveal a substantial amount of coextracted eukaryotic mRNA, when taking the bacteria to host RNA ratio into account. In general data suggest higher *16S* to *RPL32* ratios for pig feces than for chicken. Regardless of the sample type and the storage conditions, the PM kit revealed a high (except ZM buffer at 4 °C), while the NO kit revealed a low bacteria to host RNA ratio.

## DISCUSSION

In this study, we compared the impact of the storage media, temperature and length as well as RNA extraction approaches on the recovery and integrity of microbial RNA extracted from chicken and pig fecal samples. Our findings showed that the quantity and

quality of RNA varied significantly by the extraction kit. Pig feces samples showed the lowest variability and chicken feces samples exhibited the greatest variability regarding differences in the extraction method. Overall, the PM extraction kit in combination with storing the fecal samples in RL at either 4 °C (for 24 h) or −80 °C (up to 2 weeks) performed best for both chicken and pig samples considering the RNA yield, the RNA integrity and finally the level of host contamination, as an additional quality metric. The presence of host RNA, as estimated by targeting the chicken and pig *RPL32* mRNA by RT-qPCR for this study, constitutes a severe technical issue in metatranscriptomic analysis. Obtained results uncovered coextraction of host RNA in all samples, regardless of the extraction kit and storage condition, with the PM kit showing overall the highest bacterial to host RNA ratio.

Our results further underscored the importance of a standardized protocol or reporting guidelines for sample storage, duration and RNA extraction approach. This is very important as the possible effect of these technological measures can potential mask the actual biological heterogeneity to be described in the microbiome studies. Human microbiome researchers have already advocated for reporting guidelines and standardization procedures in their field (*Amos et al., 2020*; *Mirzayi et al., 2021*; *Tourlousse et al., 2021*). Developing similar standards is crucial to accelerate progress in the area of livestock microbiome research.

To the best of our knowledge, our study is the first one investigating the impact of sample storage and RNA extraction strategies on the recovery of high-quality microbial RNA for livestock gut microbiome analysis. However, there are a few studies that have been performed on human stool or biofilm samples (*Cardona et al., 2012*; *Yao, Rao & Habimana, 2021*). In 2015, *Reck et al. (2015)* compared different RNA extraction kits and RNA storage solutions for human stool metatranscriptome analyses. They compared the Chloroform:Isoamylalcohol RNA extraction protocol used by *Zoetendal et al. (2006)* with four commercially available kits, two of which were also used in our study. They found that the PM Kit performed best with respect to RNA yield and purity. Considering these parameters in our study, the PM kit was outperformed by the MN kit, which was not included in the study by *Reck et al. (2015)*. Since the stability of RNA is a crucial factor in metatranscriptome studies, *Reck et al. (2015)* also had a closer look on the effect of storage and stabilization reagents on the final RNA quality by comparing four different available RNA stabilizers over a time-period of 360 h (*Reck et al., 2015*; *Zoetendal et al., 2006*). Overall, this study provides information on the stability of human stool metatranscriptome under different preservation and storage conditions. Other studies performed in human stool samples have addressed the issue of RNA stabilization after sampling, showing that the RNALater kit was the most successful stabilizer and protector of RNA during storage, even if different RNA extraction procedures were used (*Franzosa et al., 2014*; *Seelenfreund et al., 2014*; *Song et al., 2016*). In line with these results our study suggests that RL is also the best RNA stabilizer for chicken and pig intestinal samples.

Analysis of the metatranscriptome *via* RNA sequencing is often impaired by the low abundance of prokaryotic mRNA. As the mRNA accounts only for a small subset of the total RNA, the removal of rRNA before starting library preparation has already been

implemented in various studies on the gut microbiome to avoid wasting reads in the sequencing process and allow cost-effective MTX analysis (*Faits et al., 2020*; *Ogunade, Pech-Cervantes & Schweickart, 2019*; *Reck et al., 2015*; *Sher et al., 2020*; *Wang et al., 2020*). Still, high abundance of host RNA (as *e.g.*, commonly obtained in biopsy samples or samples with low bacterial load) constitutes a technical issue: Commercially available rRNA depletion kits might not provide a sufficient high level of enrichment in bacterial mRNA, which can affect the sequencing depth for the microbiota, increase overall processing costs and complicate downstream analysis (*Bashiardes, Zilberman-Schapira & Elinav, 2016*; *Bikel et al., 2015*). Therefore, to ensure high bacteria to host RNA ratios, the application of extraction protocols revealing only low levels of host RNA contamination or methods for host RNA depletion are suggested (*Giannoukos et al., 2012*; *Kumar et al., 2016*; *Robbe-Saule et al., 2017*). Applying a protocol for differential lysis of eukaryotic cells, *Robbe-Saule et al. (2017)* were able to reduce the level of host RNA five times, compared to the initial RNA extraction protocol. The level of co-extracted host RNA contaminants, as evaluated in this study, might therefor be another important parameter in selecting appropriate microbial RNA extraction kits and constitutes a valid starting point for further optimization. Further measures of RNA depletion might be a trade of costs and personal hours added upon inclusion of further steps, but might be of special interest, when studying the *in vitro* transcriptome of bacterial pathogens. Applying a targeted RNA sequencing approach by differential cell lysis and probe-based ribosomal depletion, a 50-fold enrichment in the gene number of *Mycobacterium tuberculosis* was obtained in an *in vivo* infection model (*Cornejo-Granados et al., 2021*).

While, to the best of our knowledge, there is no information on the application of MN and ZY kit in gut microbiome research available in literature, the PM extraction kit, showing overall good performance in our study, was used successfully (with or without storing the fecal samples in RL) for metatranscriptomics on human, pig, chicken and elephant feces samples as well as on infant gut and cattle rumen samples (*Faits et al., 2020*; *Güllert et al., 2016*; *Ogunade, Pech-Cervantes & Schweickart, 2019*; *Reck et al., 2015*; *Sher et al., 2020*; *Wang et al., 2020*). Based on the available data, the PM kit might be a good candidate for RNA extraction of other monogastric livestock species as well. By contrast, we obtained low RNA yields and poor RNA quality for pig and chicken samples extracted with the NO kit, which has been used successfully for RNA-Seq analysis in a study on human gut microbiome (*Tarallo et al., 2019*). As matrix dependent effects of the storage and RNA extraction methodologies cannot be ruled out and as compared to the Metagenome based research of human, animal or environmental derived samples, only view technical studies for metatranscriptomic analysis are available, further investigation might be necessary.

In many studies fecal samples are used as a proxy to study the gut microbiota. However, these may not be fully representative of the whole gastrointestinal tract. In this study we used colon fecal samples, so our results may not apply to protocols where the recovery of high-quality RNA is performed from other chicken or pig gastrointestinal regions such as duodenum, jejunum, ileum or cecum and they may have suffered from the small size of our sampling material (one pig and eight chicken). Sample size in terms of the

number of animals was small, we cannot exclude the likelihood that the results observed here represent those from a larger number of individuals. Although there are gut microbiota studies that reveal good correlation between 16S rRNA and mRNA stability (*Reck et al., 2015*), rRNA is generally considered as more stable and might therefor only be a moderate indicator for the stability of the present mRNA. Nevertheless, our findings are a starting point in electing the most efficient combination of parameters and kits to extract high quality RNA that is representative of the chicken and pig gut microbiota.

## CONCLUSIONS

In this study, we highlighted how key stages of conducting a metatranscriptomics study on livestock microbiome, including the sample storage and RNA extraction, can potentially affect the results and therefore their possible biological interpretation. Our results suggest that the shipping of chicken and pig fecal samples on normal ice (4 °C) from farm to laboratory within 24 h, while stabilized in RNALater buffer, followed by extraction with Rneasy Power Microbiome kit could be established as a standard practice for large cohort livestock microbiome studies.

## ACKNOWLEDGEMENTS

We thank Nikolaus Grabner and Andreas Köstelbauer for their outstanding technical assistance.

### Funding

This research was funded by the Austrian Research Promotion Agency (FFG) through the projects "Frontrunner: Omics technologies and natural feed additives-solving challenges of livestock industry in the area of digitalization" (Project Number 866384) and COMET-K1 Competence Centre for Feed and Food Quality, Safety and Innovation (FFoQSI GmbH, Project Number 854182). The COMET-K1 competence centre FFoQSI is funded by the Austrian ministries BMVIT and BMDW and the Austrian provinces Niederoesterreich, Upper Austria, and Vienna within the scope of COMET—Competence Centers for Excellent Technologies. The program COMET is handled by the Austrian Research Promotion Agency FFG. The funders had no role in study design, data collection and analysis, decision to publish, or preparation of the manuscript.

### Grant Disclosures

The following grant information was disclosed by the authors:
Austrian Research Promotion Agency (FFG): 866384.
COMET-K1 Competence Centre for Feed and Food Quality, Safety and Innovation (FFoQSI GmbH): 854182.
Austrian Research Promotion Agency FFG.

## Competing Interests

Mahdi Ghanbari and Gertrude Wegl are employed by Biomin Holding GmbH.
The authors declare that they have no competing interests.

## Author Contributions

- Raju Koorakula conceived and designed the experiments, performed the experiments, analyzed the data, prepared figures and/or tables, authored or reviewed drafts of the article, writing—original draft preparation, and approved the final draft.
- Mahdi Ghanbari conceived and designed the experiments, analyzed the data, authored or reviewed drafts of the article, writing—original draft preparation,supervision, investigation, resources,funding acquisition, and approved the final draft.
- Matteo Schiavinato conceived and designed the experiments, authored or reviewed drafts of the article, supervision, and approved the final draft.
- Gertrude Wegl conceived and designed the experiments, prepared figures and/or tables, authored or reviewed drafts of the article, supervision, investigation, resources, and approved the final draft.
- Juliane C. Dohm conceived and designed the experiments, authored or reviewed drafts of the article, supervision,resources, and approved the final draft.
- Konrad J. Domig conceived and designed the experiments, authored or reviewed drafts of the article, supervision, resources,funding acquisition, and approved the final draft.

## Animal Ethics

The following information was supplied relating to ethical approvals (*i.e.*, approving body and any reference numbers):

This animal study not required ethical review and approval because samples from pig and chicken were collected from conventional slaughterhouses.

## Field Study Permissions

The following information was supplied relating to field study approvals (*i.e.*, approving body and any reference numbers):

This animal study not required permission from any institution because samples from pig and chicken were collected from conventional slaughterhouses.

## Data Availability

The raw data is available in the Supplemental Tables 1 and 2.

## Supplemental Information

Supplemental information for this article can be found online at http://dx.doi.org/10.7717/peerj.13547#supplemental-information.

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
