# Peer review of "Storage media and RNA extraction approaches substantially influence the recovery and integrity of livestock fecal microbial RNA"

_PeerJ, doi:10.7717/peerj.13547_

## Round 0.1 · original submission · Major Revisions

Please provide a comprehensively revised version addressing the editorial comments and a detailed rebuttal letter.

Please address the concerns in particular the size of the sample (n) and the location of where the sampling occurred.

Reviewer 1 ·

Basic reporting

The study by Koorakula et al. aims at assessing and comparing the impact of different storage and extraction methods for the isolation of microbial RNA from fecal samples. The study is clear and concise, with an adequate experimental design for the intended objective. However, the observations and conclusions reached are limited and do not seem to contribute to the existing knowledge on the field.
Although few studies have been dedicated to comparing RNA extraction methods exclusively, many of the statements in Koorakula et al. have already been reported in other studies (doi.org/10.3389/fmicb.2021.588025, doi.org/10.1186/1471-2180-12-158).

Experimental design

The extraction of sufficient and good-quality RNA is essential for metagenomic studies, and finding the best extraction protocol for each sample type is a question every research group has to explore.
The experimental design in this study is overall adequate for the intended objective. However, the conclusions reached have already been suggested in other articles.

Validity of the findings

The notion that RNA extraction kits and sample origin impact the quality and quantity of recovered RNA has already been addressed in metagenomic reports.
However, the effort done in this study would benefit from additional exploration of the RNA obtained with the different methods.
For example, performing a qPCR to determine the amount of host RNA remaining in the sample. This is a very important question when intending to analyze metagenomic data

Additional comments

The study is well designed and answers the proposed objective.
Consider exploring further the characteristics of the RNA obtained by the different methods to add value to the observations and conclusions. For example, exploring the amount of host RNA or if the perceived proportion of any taxa changed would greatly benefit the study.
Also, when mentioning the amount of RNA recovered, mention the total amount in ng, not only the concentration (ng/µL), and specify the quantification method you are referring to.

Reviewer 2 ·

Basic reporting

The manuscript by Koorakula et al, aimed to determine the impact of sample storage and RNA extraction on the recovery of high-quality RNA.

Experimental design

The experimental design lacks samples, the authors only collect 1 pig sample and 8 chicken sample that were pooled by animal to perform further analysis. It is not clear why they call it fecal samples when they were collected from the distal part of the large intestine. The description of sample collection is not appropriate. Usually, samples used for RNA extraction should be collected and stored directly after slaughtering without moving them to the laboratory and proceeding with the storage.
The authors only share the minimum information about RNA concentration and integrity. It is missing information regarding 260/280 and 260/230 rations. In the supplementary table, we see values for RNA yield but no idea if they were measure with qubit or nanodrop.

Validity of the findings

This study lacks a critical experiment that would answer the question, is this RNA good enough to perform downstream analysis, is it sufficient to run a qPCR or RNA-seq? This should be answered here to prove their conclusions regarding the kit of choice.

·

Basic reporting

The authors have compared the efficiency of RNA recovery and integrity for four storage conditions (2 media X 2 temperatures-length) of primary samples of pig and chicken feces followed by RNA extraction using one of four commercial kits.

The writing is generally clear enough to ensure that an international audience can understand the text. However, I have noted a number of errors in structure, especially in the Conclusion section, comma usage and of choice of prepositions (e.g. in, for), some of which I have detailed bellow. I suggest the text should be revised by someone proficient in the English language for improvement.

The submission is «self-contained». The structure of the article conforms to the journal’s standard. Figures are relevant, of sufficient resolution and appropriately labeled. Raw data is provided.

Specific comments / corrections

Lines 31- 35. I suggest reordering the sentences to follow chronology:
« Fecal samples from pig and chicken were collected from conventional
slaughterhouses. Two different storage buffers were used at two different storage
temperatures. Extraction of total RNA was done using four different commercially
available kits and RNA integrity/quality and concentration were measured using a
Bioanalyzer 2100 system with RNA 6000 Nano kit (Agilent). »

Line 51. I suggest replacing by «… broken down through enzyme production by the animal host, and in …»

Line 54. I suggest replacing by «… in terms of taxonomy and phylogeny..»

Line 55-56. I suggest replacing by «… subunit gene, of which either the full length or its hypervariable regions are targeted for sequencing and used for taxonomic classification.»

Line 57-58. I find using the terms « Whole genome shotgun sequencing » may be confusing as « whole genome sequencing » generally refers to sequencing of a bacterial isolate genome, while « shotgun » as in « shotgun metagenomic sequencing » refers to sequencing all the DNA present in a primary sample. Therefore, I suggest using « shotgun metagenomic sequencing ».

Line 74. I suggest replacing by «… actually active in a given context..»

Line 89. I suggest using «next generation sequencing» instead of NGS as it was not previously defined and is only used once in the whole manuscript.

Line 93. …matrices…

Line 95 … are performed on fecal samples…

Line 96 … to as ZM) used in two storage conditions (overnight at 4C or 2 weeks at -80C), and in combination with…

Line 105. « Fecal sample from a 180 days old pig »

Line 109. I am not sure what is meant by « the respective gut digesta ». If there is only one pig sample, how was it pooled? If I interpret right the authors writing, I suggest the following modification instead of the sentence found in lines 107-112: «…transported to the laboratory. Samples from the digesta in the distal part of the large intestine, herein referred to as feces, were gathered. The pig fecal sample, as well as the pooled fecal chicken samples, were homogenized and then transferred...»

Line 116. Add references to support «the freezing period was considered of minor importance» statement.

Line 194. «followed by a» is duplicated.

Line 211-212. «…stored in RL for two weeks at -80C and extracted with the MN kit….»

Line 226. Why is « pig feces samples » plural if there was only one (n=1, line 105)?

Line 235. «…storage in ZM buffer… and extraction using the ZY kit…»

Line 238. Kit, singular

Line 243. «…storage in RL…»

Line 245-248. Poor English, please rephrase.

Line 250. Capital U in «Unlike»

Line 255. «… we compared…»; «storage media, temperature and length, as well as..»

Line 257. «…Our findings...»

Lines 257-260. Poor English, please rephrase.

Lines 263-264. I suggest specifying the scope of this statement. I do not think it is realistic that all the people doing microbiome studies worldwide use the same sample storage and extraction method, but I totally agree with the authors that results can only be compared when the samples were processed using exactly the same methods and conditions.

Lines 272-279. Did Reck et al. use the same methods as the authors? Compare their findings with those of this study.

Lines 283-284. Again, compared the findings of these previously published studies with your findings.

Lines 286-291. Extend the chicken focus to include pigs.

Lines 291-293. Rephrase: microbiome profiles were not investigated in this study.

Line 295. What about the pig gut microbiota?

Line 299. Capital W in «We»

Experimental design

This work fits within the Aims and Scope of the journal. The research question is well defined, relevant and meaningful. A knowledge gap is filled. However, the methods should be described with more clarity. Specifically, in lines 119-171, the only adjustments made to the manufacturer’s protocols were the use of 150 mg of samples and the volume of the elution buffet? Then, lines 134-171 are only summaries of the manufacturer’s protocols steps? The level of details provided for each methods varies. Were the bead beating conditions for each methods determined by the manufacturers? To ensure a good comprehension for the reader, I suggest revising this section to put clearly forward what were (all) the adjustments made for each method, and aim at providing equivalent levels of details for each method.
The details of the samples (lines 137 – 140) should be in the « RNA extraction » section as it applies to all the extraction methods. Moreover, the granularity provided should include the two animal types.

Line 174. If I understand right, there was a total of 96 individual extractions, not 48:
2 animal types X 2 stabilization reagent X 2 storage conditions X 4 extraction methods X triplicates = 96 extractions

The authors work provides important stepping stones for studies involving metatranscriptomics of livestock feces, but I am finding the experimental set up a bit thin. First, I find the inclusion of only one pig sample (line 105) very limited. It is hardly representative of what can be encountered in pig fecal samples that can vary substantially depending on the animal age, health status, diet and husbandry practices. Second, I suggest either rephrasing or additional experiments to support the conclusions/statements found in lines 264-266, « This is very important as the possible effect of these technological measures is high relative to the actual biological heterogeneity to be described in the microbiome studies. », line 299 «affects the results and therefore their possible biological interpretation» and line 303 «reliable microbial RNA extraction». Although purity ratios are important indicators of sample quality, the best indicator of DNA or RNA quality is functionality in the downstream application of interest. RNA Integrity as observed by RIN is one thing, but an assay showing the actual quantification/sequencing of targets of interest would consolidate the work. I am suggesting assays of qRT-PCR targeting a number of genes (e.g. 3, one of which could be the 16S rRNA gene, others could be target genes known to be present and expressed at different levels in the studied microbiome). Results of all, or a selected subsets, of the RNA preparations could be compared to extrapolate on the storage & extraction methods robustness.

Validity of the findings

The underlying data have been provided. They appear robust and statistically sound.
Some conclusions are overstepping the limits of the supporting results, see the additional experiments I am proposing in the «Experimental design» section of my review.

---

## Round 0.2 · Major Revisions

Please provide a comprehensively revised version addressing the editorial comments and a detailed rebuttal letter.

Reviewer 1 ·

Basic reporting

The authors have improved the manuscript by addressing the reviewer's concerns. However, the manuscript needs a broader discussion about their observations and rationale for this study, contrasting them with previous studies. These will define the study better for the readers.

Experimental design

The experimental design in this study is overall adequate for the intended objective. In addition, the authors included the suggested qPCR experiments providing a better notion about the RNA quality. Please, include these observations in the discussion section and mention the relevance of these results for someone who wants to perform high throughput sequencing techniques such as RNA-seq

Validity of the findings

I understand the author's rationale that most studies have been performed to research human and environmental samples, and these findings cannot be directly transferred to the animal microbiota. Thus, consider including a paragraph in the discussion section on whether or not these results can be transferred to livestock animals other than chicken and swine. Furthermore, expand the comparison between this study's observations and the results observed in humans. These can help clarify the relevance of the study.

---

## Round 0.3 · accepted · Accept

Thanks for addressing all the revisions and corrections requested. Now your manuscript is accepted in PeerJ.

Reviewer 1 ·

Basic reporting

The authors have improved the manuscript by addressing the reviewer's concerns. They have enriched the discussion including key observations.

Experimental design

The experimental design is clear and adequate for the intended objective

Validity of the findings

The authors have enriched the discussion and conclusions supporting the results.

Additional comments

No additional comments